# Neutrophils Actively Contribute to Obesity-Associated Inflammation and Pathological Complications

**DOI:** 10.3390/cells11121883

**Published:** 2022-06-10

**Authors:** Eileen Uribe-Querol, Carlos Rosales

**Affiliations:** 1Laboratorio de Biología del Desarrollo, División de Estudios de Posgrado e Investigación, Facultad de Odontología, Universidad Nacional Autónoma de México, Mexico City 04510, Mexico; euquerol@fo.odonto.unam.mx; 2Departamento de Inmunología, Instituto de Investigaciones Biomédicas, Universidad Nacional Autónoma de México, Mexico City 04510, Mexico

**Keywords:** neutrophil, obesity, adipose tissue, inflammation, diabetes, neutrophil-to-lymphocyte ratio, macrophage, microbiota

## Abstract

Obesity is characterized by an increase in body weight associated with an exaggerated enlargement of the adipose tissue. Obesity has serious negative effects because it is associated with multiple pathological complications such as type 2 diabetes mellitus, cardiovascular diseases, cancer, and COVID-19. Nowadays, 39% of the world population is obese or overweight, making obesity the 21st century epidemic. Obesity is also characterized by a mild, chronic, systemic inflammation. Accumulation of fat in adipose tissue causes stress and malfunction of adipocytes, which then initiate inflammation. Next, adipose tissue is infiltrated by cells of the innate immune system. Recently, it has become evident that neutrophils, the most abundant leukocytes in blood, are the first immune cells infiltrating the adipose tissue. Neutrophils then get activated and release inflammatory factors that recruit macrophages and other immune cells. These immune cells, in turn, perpetuate the inflammation state by producing cytokines and chemokines that can reach other parts of the body, creating a systemic inflammatory condition. In this review, we described the recent findings on the role of neutrophils during obesity and the initiation of inflammation. In addition, we discuss the involvement of neutrophils in the generation of obesity-related complications using diabetes as a prime example.

## 1. Introduction

Obesity is a clinical condition defined as a lopsided body weight associated with an exaggerated enlargement of the adipose tissue [1,2]. In simple terms, obesity appears when people have low physical activity (a sedentary lifestyle) and increased ingestion of food, particularly of high-energy-yielding groceries [3]. However, obesity is more complex than an imbalance between caloric intake and energy requirements. External factors, such as genetics, socioeconomic status, and environment, influence food consumption, nutrient assimilation, thermogenesis, and fat storage in various adipose tissues [4]. At the present time, 39% of the world population is obese or overweight. The prevalence of these conditions has steadily increased globally during the last 30 years [5], and it is expected that, if this tendency continues, by 2025, obesity prevalence in the world will be 18% in men and 21% in women [6]. This tendency makes obesity the 21st century epidemic [7,8] with an important economic burden on health costs because obesity is associated with the development of multiple pathological conditions also known as obesity-related complications, including type 2 diabetes mellitus, cardiovascular diseases, some types of cancer, and coronavirus disease 2019 (COVID-19) [3,9,10,11].

Obesity is also characterized by a mild, chronic, systemic inflammation. The excessive accumulation of fat in the adipose tissue is due to either an increase in adipocyte size (hypertrophy) or the growth of new adipocytes (hyperplasia). Both conditions cause adipocyte stress and malfunction leading to inflammation [12]. As obesity worsens, the initial mild (low-grade) inflammation becomes chronic, and later it turns systemic [13,14,15]. The adipose tissue is composed not only of adipocytes (mature fat cells), but also of different types of immune cells, which are important for tissue homeostasis [16,17]. During obesity, stressed adipocytes produce adipokines, which recruit and activate innate immune cells [18]. These immune cells in turn perpetuate the inflammation state via production of cytokines and chemokines that can even influence other parts of the body, creating a systemic inflammatory condition [19,20].

The vast majority of immune cells infiltrating the adipose tissue are macrophages [21,22]. These immune cells accumulate around adipocytes in crown-like structures [23] where macrophages can proliferate and further maintain the adipose tissue inflammation [24]. However, recently it has become evident that neutrophils, the most abundant leukocytes in blood and the primary effector cells of acute inflammation, are also the first immune cells infiltrating the adipose tissue [25]. Neutrophils then get activated [26,27,28] and release multiple inflammatory factors and chemokines that recruit macrophages and other immune cells including B cells, T cells, and NK cells [29]. The crosstalk between adipocytes and neutrophils further promotes the adipose tissue inflammation [30]. In this review, we describe the recent findings on the role of neutrophils during obesity and the initiation of a low-grade systemic inflammation. In addition, we discuss the involvement of neutrophils in the generation of obesity-related complications using diabetes as a prime example.

## 2. Obesity

Obesity is associated not only with an excessive accumulation of fat in the adipose tissue [1,2], but also with a mild, chronic, systemic inflammation [12]. Increased fat accumulation results from an imbalance between the energy derived from food ingestion and the energy used for body functions. However, this simple view (based on thermodynamics) does not take into consideration the complexity of body weight control. Body weight is influenced not only by the quality of the diet and the amount of physical activity, but also by work schedules, ambient temperature, lack of sleep; drugs that modify endocrine and reproductive functions, gut microbiota, and epigenetic effects [4]. The complexity of weight control makes the etiology of obesity an intricate mix of personal choices, socioeconomic level, and environmental factors [2]. As a result, obesity has become a pandemic health problem [7,8]. In 2016, the World Health Organization (WHO) estimated that more than 1.9 billion people, 18 years and older, were overweight. Of these, over 650 million were obese. In addition, over 340 million children and adolescents aged 5–19 were also overweight or obese [31].

Assessment of obesity is not easy since there are multiple ways to accumulate fat in tissues. This variation creates two principal types of obesity: subcutaneous and visceral. In subcutaneous obesity (which is more common in women), excess fat is found under the skin surrounding the hip and thigh areas, while in visceral obesity (which is more common in men), fat is concentrated in the abdominal region, primarily in the mesenteric adipose tissue. In order to assess obesity, several indicators have been used. The body mass index (BMI) is one such indicator that estimates overweight and obesity based on the weight of the individual expressed in kilograms (kg) divided by the square height of the individual in meters (m^2^) [31]. In addition, other indicators besides the BMI should be considered to fully assess the severity of obesity. For example, waist circumference helps to discriminate between subcutaneous obesity and visceral obesity, and it is a good indicator of poor health. A strong correlation of waist circumference with all-cause mortality was found for men (with waist circumferences larger than 110 cm) and for women (with waist circumferences larger than 95 cm) [32].

Obese individuals are at a greater risk of developing numerous health problems that play a role in premature death [5,31,32], including features of the metabolic syndrome [1,33], and other obesity-related complications such as type 2 diabetes mellitus, cardiovascular diseases, some types of cancer [3], and COVID-19 [11]. In addition, obesity has a negative influence on psychological and cognitive functions [34,35].

## 3. Microbiota and Obesity

Obesity is usually accompanied by a mild, chronic, systemic inflammation. This inflammatory state is initiated by damage to the adipose tissue due to excessive fat deposition (see later), and also by increased intestinal permeability due to alterations to the gut microbiota [36]. The gut microbiota is the community of microorganisms (including bacteria, archaea, fungi, protozoa, and viruses) that reside in the digestive tract. The dominant phyla in the gut are Firmicutes (Gram-positive) (60–65%), Bacteroidetes (Gram-negative) (20–25%), Proteobacteria (such as *Escherichia* and Enterobacteriaceae) (10%), and Actinobacteria (Gram-positive) (3%) [37,38].

Recent years have seen an increasing evidence of the association of changes in gut microbiota with the development of obesity (reviewed in [39,40,41,42,43]). This association was evidenced through initial reports showing that germ-free mice fed a high-fat diet had reduced body fat and did not develop obesity [44,45]. However, these mice recuperated fat and developed insulin resistance and glucose intolerance after reconstitution with the gut microbiota from normal mice [45]. Then, a metagenomic analysis showed that the two main bacterial phyla, Firmicutes and Bacteroidetes [37], maintain a relatively constant balance both in lean mice and humans; but in obese individuals, higher levels of Firmicutes and lower levels of Bacteroidetes were found [46,47]. Interestingly, after eliminating the high-fat diet, the ratio of Firmicutes/Bacteroidetes was reverted to the levels found in lean individuals [46]. Most other studies have had similar results [40]. However, numerous reports of opposite results suggest that the Firmicutes/Bacteroidetes ratio not always increases in obesity, and thus, further studies are required to resolve this issue [48]. Nevertheless, it seems that changes in specific bacteria, more than the ratio of the phyla, are better associated with obesity [49].

Altering the gut microbiota composition has important metabolic consequences. For example, obesity-associated microbiotas have been found to be more efficient at obtaining energy from the diet by producing enzymes that degrade food more efficiently, thus favoring a larger caloric intake [47]. Another significant consequence of an altered microbiota during obesity is the increased intestinal permeability [50], which leads to the escape of bacteria and bacterial products across the intestinal barrier into the blood circulation [36]. A major bacterial product found to cross the intestinal barrier in obese individuals is lipopolysaccharide (LPS), which triggers innate immune cells, leading to inflammation [51,52]. A high-fat diet increases the growth of Gram-negative bacteria and promotes LPS absorption across the intestinal barrier [51,53]. Thus, LPS absorption caused by ingestion of high-fat diets is related to obesity-induced low-grade systemic inflammation [54].

In contrast, other bacterial products such as short-chain fatty acids (SCFAs) have been associated with maintenance of reduced weight [55]. SCFAs, predominantly butyrate, acetate, and propionate, are produced by fermentation of nondigestible carbohydrates (dietary fiber) by various gut bacteria [56] and serve as an energy source for the colonic epithelia (butyrate), liver (propionate), and peripheral tissues (acetate) [57]. In particular, butyrate and propionate induce the production of gut hormones, such as hormone peptide tyrosine (PYY) and glucagon-like peptide-1 (GLP-1) from colonic L cells [58] via their cognate free fatty acid receptors (FFARs) [59]. In this way, SCFAs lead to reduced food intake, resulting in weight loss and the maintenance of reduced weight [50,59,60]. In addition, SCFAs can diminish inflammation by modulating several leukocyte functions, including reduced production of inflammatory cytokines and eicosanoids, inhibition of leukocyte migration to inflammation sites [61], and induction of neutrophil apoptosis [62].

Based on the evidence that microbiota composition can modulate body weight and inflammation, a growing interest exists to modify the gut microbiota by different approaches. The use of probiotics (live microorganisms which in proper quantities could provide a health benefit) or of prebiotics (nondigestible food ingredients that favor growth of health-promoting bacteria) has been extensively tried [63,64,65]. Furthermore, fecal microbiota transplantation is now considered as a feasible therapy for treating obesity by influencing the composition of the gut microbiota [66]. Although some reports suggest that these approaches can reduce obesity and other associated complications [41,67], in general, only a limited success is achieved by microbiota-related interventions for body weight control in humans [68]. Therefore, future research is needed combining different approaches to improve the beneficial effects of microbiota modification on obesity.

Nevertheless, it is clear that an altered microbiota during obesity can induce a low-grade systemic inflammation by increasing intestinal permeability [50] and allowing bacterial products, such as LPS, to cross the intestinal barrier into the blood circulation [36].

## 4. Changes of the Adipose Tissue during Obesity

There are two principal types of adipose tissue: brown adipose tissue and white adipose tissue. Together, these tissues are involved in maintaining the energy balance of the body [69]. Adipocytes (adipose cells) in brown adipose tissue are involved in the regulation of body temperature by generating heat (thermogenesis) using lipid reserves. This tissue is abundant in newborn babies and is found in the interscapular and supraclavicular regions, and around critical organs such as the heart, the kidneys, the trachea, and the pancreas. It is believed that this anatomical distribution of brown adipose tissue is important for protecting vital organs from hypothermia [70]. Brown adipose tissue is present not only in newborns, but also in adult humans, and its mass correlates with leanness [71]. White adipose tissue is the most abundant one throughout the body of adult humans. The main function of this tissue is the storage of energy in the form of lipids (triglycerides). However, recent studies have shown that adipocytes not only accumulate fat, but also are capable of secreting bioactive substances, collectively known as adipocytokines or adipokines [72,73]. In addition, a third type of adipose tissue is also found within white adipose tissue. This minor type is known as “beige” or “brite” adipose tissue, and it is also involved in thermogenesis [69,71]. As mentioned before, adipose tissues are also classified according to their location in the body as subcutaneous or visceral. Visceral adipose tissue is more cellular (adipocytes) and vascular than subcutaneous tissue and contains more immune cells (see later) [74]. Visceral adipose tissue is of particular interest because accumulation of fat in this tissue correlates with the development of the metabolic syndrome [75].

During obesity, the principal change observed in adipose tissue is the accumulation of fat. As a result of energy surplus, adipose tissues must store more lipids. This is achieved by first inducing adipose tissue hyperplasia. The increased number of adipocytes expands the tissue, which then serves as a “metabolic sink” distributing the excess fat in the growing tissue. Later, when the tissue cannot expand anymore (due to causes that are not well-identified), stored lipids (triglycerides) induce adipocyte hypertrophy. Adipocytes enlarge until they become saturated and are no longer functional [2] (Figure 1).

In addition, when adipose tissue has no more room for storing the excess of triglycerides, an abnormal deposition of lipids begins to appear in other normally lean tissues, such as the heart, the liver, and the kidneys. This event is known as ectopic fat deposition [75], and it has harmful effects on metabolism, including insulin sensitivity and dyslipidemia (high levels of triglycerides or low levels of high-density lipoprotein cholesterol) [76,77]. Interestingly, visceral obesity is more frequently associated with ectopic fat deposition than subcutaneous obesity [78].

Adipocytes with excessive lipid storage become severely stressed. The increase in size is tolerated up to a certain (threshold) point, beyond which adipocytes develop molecular alterations resulting in dysfunction and death [79]. Interestingly, visceral adipocytes become dysfunctional at a smaller size than subcutaneous adipocytes [21]. This fact may explain why visceral obesity is clearly associated with obesity-related complications. An important molecular alteration observed in dysfunctional adipocytes is the endoplasmic reticulum (ER) stress response, which is activated by incorrectly folded proteins within the ER [80], and it is connected to inflammatory pathways [81]. Another molecular alteration observed in stressed adipocytes is the response to hypoxia. As adiposity becomes severe, lower levels of oxygen are present in adipose tissues. This results in activation of the hypoxia-inducible factor (HIF) signaling cascade, which in turn upregulates proinflammatory responses [82]. Therefore, stressed adipocytes can release large amounts of proinflammatory adipokines, such as leptin [83], and cytokines, such as tumor necrosis factor alpha (TNF-α), interleukin (IL) 6, monocyte chemoattractant protein 1 (MCP-1), IL-1, and IL-8, which further promote inflammation [72,73]. All these changes create a low-grade inflammatory state in obese adipose tissue that is further amplified by cells of the immune system.

Macrophages are an abundant population of immune cells in adipose tissue. Some macrophages exist in lean adipose tissue, but their numbers increase dramatically during obesity. In obese adipose tissue, macrophages concentrate around apoptotic and dead adipocytes, forming crown-like structures where defective adipocytes are eliminated by phagocytosis [23] (Figure 1b). The increase in macrophage numbers is due to a combination of recruitment, retention, and proliferation of these cells. Recruitment is mediated by myeloid C–C motif chemokine receptor-2 (CCR2), the receptor for MCP-1 [22], although other chemotactic agents also participate in macrophage recruitment to the adipose tissue [84]. Macrophage retention entails the direct contact of macrophages with adipocytes. This cell–cell contact is mediated by adhesion of integrin α4β1 on macrophages with vascular cell adhesion molecule-1 (VCAM-1) on adipocytes [85] (Figure 2). Macrophages are also capable of proliferating, particularly in the crown-like structures [24] where the inflammatory microenvironment provides Th2 cytokines, such as IL-4, IL-13, and granulocyte/monocyte colony-stimulating factor (GM-CSF), that stimulate macrophage growth [86].

Accumulation of macrophages in obese adipose tissue goes together with the activation of these cells. Activation results in an increase in macrophages with the M1 (proinflammatory) phenotype and a decrease in M2 (anti-inflammatory) macrophages [22,87,88]. The proinflammatory activation of macrophages in the adipose tissue involves many different extracellular factors, including lipids, adipokines, and cytokines [23,89,90]. Lipids, such as free fatty acids, deriving from enhanced lipolysis of dysfunctional adipocytes, are capable of activating macrophages via the interaction with toll-like receptor 4 (TLR-4) [90,91], resulting in the activation of proinflammatory signaling pathways, like the nuclear factor kappa B (NF-κB) and c-Jun N-terminal kinase (JNK) pathways [16]. Mechanistically, free fatty acids, however, do not directly engage TLR-4. Instead, they bind fetuin-A, which then binds to TLR-4, leading to the activation of proinflammatory signals [92,93] (Figure 2). Adipokines such as leptin, the hormone that inhibits hunger, are important activators of macrophages. Leptin induces macrophages to produce TNF-α, IL-6, MCP-1, and IL-1β [83,94]. Cytokines such as TNF-α, IL-6, and IL-1β are also important activators of macrophages [20,95]. Because all these proinflammatory factors are produced either by dysfunctional adipocytes [20] or by activated macrophages themselves, the activation of macrophages in obese adipose tissue becomes self-sustained. This is probably one of the reasons for the continuous low-grade inflammation of obesity [9,96].

So far, most studies on obesity have focused on the role of activation of macrophages and their accumulation in adipose tissues (reviewed in [89,95,97]). However, many other immune cells are also found in adipose tissues, including neutrophils, CD4 T cells, CD8 T cells, B cells, dendritic cells (DCs), and mast cells [17,98]. Recently, it has become evident that neutrophils are the first innate immune cells infiltrating obese adipose tissues and important regulators of the obesity-associated inflammation. We describe next the novel findings on the role of neutrophils in obesity.

## 5. Neutrophils in Obesity

### 5.1. Circulating Neutrophils Increase in Obesity

Neutrophils are the most abundant leukocytes in human blood, the primary effector cells of acute inflammation and the first responders to infections [99,100]. Neutrophils are typically considered to be the major leukocytes against infections due to their capacity to act as phagocytic cells [101], degranulate releasing lytic enzymes, perform an oxidative burst producing reactive oxygen species (ROS), and produce neutrophil extracellular traps (NETs) with antimicrobial potential [102,103]. Neutrophils are also considered as the main effector cells of acute inflammatory reactions since they are the first leukocytes to be recruited to inflammation sites where they are capable of producing large quantities of cytokines and chemokines including TNF-α, IL-1β, IL-8, and MCP-1 [104]. Consequently, neutrophils induce the second wave of immune cells, such as macrophages and lymphocytes, to inflammation sites [105,106].

Conspicuously, circulating neutrophils are increased in obesity [107,108,109,110,111], with a clear association between the level of neutrophil blood counts and the higher BMI [107,108,110]. Furthermore, overweight individuals with neutrophilia presented elevated serum C-reactive protein (CRP) concentrations and larger waist circumferences [107,108]. In addition, neutrophil counts were significantly higher in individuals with metabolic syndrome than in lean individuals [108]. In animal models, neutrophils have also been found to be elevated in blood vessels and infiltrating adipose tissue and the endothelium at atherosclerotic lesions [112]. Moreover, neutrophils in obese individuals present an activated phenotype as indicated by elevated plasma concentrations of myeloperoxidase (MPO) and neutrophil elastase (NE) [26,113], as well as an increased expression of CD66b, a marker of neutrophil degranulation [26,27]. Activation of neutrophils from obese individuals was also indicated by stimulation of the NF-κB signaling pathway [27] and by a higher ROS generation and enhanced release of proinflammatory cytokines [114].

Importantly, weight loss following gastric band surgery resulted in a decrease in neutrophil blood counts [115] and in proinflammatory activities of peripheral blood neutrophils [114]. These results suggest that the inflammatory condition of obesity also leads to the expansion of neutrophils [9,13,94]. In fact, elevated concentrations of acute-phase proteins have been reported in obese individuals [14,116]. As mentioned before, the adipose tissue in obese individuals is capable of producing increased levels of mediators of inflammation like TNF-α, IL-1β, IL-6, and IL-8 [117,118,119,120,121,122]. These inflammatory mediators increase bone marrow granulopoiesis [123,124,125], releasing neutrophils from the bone marrow to the peripheral circulation. Moreover, these inflammatory mediators induce de-margination of neutrophils from endothelial walls, resulting in neutrophilia [126]. In addition, the adipose tissue also produces leptin which has been shown to promote hematopoiesis [127,128,129]. Leptin can additionally stimulate the oxidative burst of neutrophils, induce chemotaxis, and inhibit apoptosis in these cells [127]. Together, these reports support the notion that, indeed, obese adipose tissue is responsible for promoting a systemic inflammation that results in the generation, increase in numbers, and activation of neutrophils. In consequence, these leukocytes are the first cells to infiltrate adipose tissues.

### 5.2. Neutrophil-to-Lymphocyte Ratio (NLR)

The clear connection between obesity and an elevated neutrophil blood count has motivated people to look for simple biomarkers of obesity and inflammation. The hematological parameter for systemic inflammation known as the neutrophil-to-lymphocyte ratio (NLR) is an easy biomarker of immune response to various infectious and noninfectious stimuli [130]. The NLR is commonly used in many medical areas as an indicator of dynamic changes of neutrophils and lymphocytes in blood during systemic inflammation. The NLR reflects the relationship between innate (neutrophils) and adaptive (lymphocytes) immune responses in various pathological conditions. Because the NLR correlates with CRP concentrations, it becomes a simple cost-effective biomarker for the detection of subclinical inflammation [131].

Accordingly, the NLR has been found to be significantly higher in obese individuals than in healthy lean individuals [132,133,134,135], with a positive correlation to the BMI [132,135,136,137,138]. As expected, the NLR is also associated with higher plasma CRP concentrations [111,132,139,140]. The same trend has been reported in mice fed with an obesogenic diet. The effect on the NLR seems to be due to changes in the gut microbiota, which affects blood leukocyte numbers [134]. The significant association between obesity and a high NLR (higher than 4) was a good predictor of increased breast cancer risk. Patients with a high NLR and a high BMI also had the worst disease-free survival [133,138]. More importantly, NLR values were found to correlate significantly with the degree of abdominal obesity [141,142]. Furthermore, in morbid obese patients, a high NLR was reported to be a powerful and independent predictor of type 2 diabetes mellitus (T2D) [143]. Hence, the NLR is a simple and accessible biomarker that provides information about the inflammatory state of obese individuals. Importantly, the NLR seems to be able to identify an ongoing systemic inflammation in overweight individuals that otherwise appear healthy [139,140]. Therefore, a higher NLR in overweight individuals may reflect the subclinical inflammation already present in this group of people.

### 5.3. Neutrophil Infiltration into Adipose Tissue

During obesity-induced inflammation in animal models, neutrophil numbers increase in the peripheral circulation. From there, neutrophils can infiltrate the adipose tissue [25] and blood vessel endothelium [112]. This suggested that neutrophils have an important role at the early stages of obesity by infiltrating the abdominal adipose tissue. Neutrophils are found in the adipose tissue of lean mice in very small numbers, approximately 1% of all immune cells in the adipose tissue [144]. Yet, in mice fed a high-fat diet, a 20-fold increase in adipose tissue neutrophils was observed as early as three days after initiation of the diet [25]. In contrast, macrophage infiltration can be detected after 7 days of a high-fat diet [145,146]. These results indicated that neutrophils are the first immune cells to be recruited into adipose tissues. Neutrophil infiltration was first described as transient because after an initial remarkable increase, the neutrophil numbers decreased [25]. However, the neutrophil numbers were higher for up to 12 weeks in the adipose tissue of the mice fed with a high-fat diet than in the adipose tissue of the mice fed a normal diet [25]. Moreover, it was later shown that early recruitment of neutrophils could be prolonged over 90 days with a constant high-fat diet [19,147]. Hence, in the early phases of adipocyte disfunction, the initial inflammatory response is characterized by neutrophil infiltration into adipose tissues.

Neutrophils are recruited to the adipose tissue via the action of several chemotactic factors produced in the obese adipose tissue. Inflamed adipocytes produce larger amounts of IL-8, a potent neutrophil chemoattractant [19,117]. Once in the adipose tissue, neutrophils can recruit more blood neutrophils by releasing C–X–C motif chemokine ligand 2 (CXCL2), another important neutrophil chemoattractant [148] (Figure 3). Furthermore, lipids extracted from human adipocytes were shown to induce migration of neutrophils and macrophages, and also secretion of other cytokines [149]. In addition, free fatty acids derived from adipocyte lipolysis could also attract neutrophils and stimulate them to produce more IL-1β, which in turn activates other adipocytes and immune cells [30] (Figure 3). The exact molecular nature of the various lipid chemotactic factors is not yet known. Future studies will help elucidate these chemotactic factors and the mechanisms they use to recruit neutrophils into adipose tissues.

### 5.4. Neutrophil Activation and Inflammation

Once in the adipose tissue, neutrophils interact with adipocytes via the binding of integrin α_M_β2 (Mac-1) on the neutrophil to intercellular adhesion molecule 1 (ICAM-1) on the adipocyte [25] (Figure 3). This interaction activates neutrophils and induces them to produce IL-1β and TNF-α, which further stimulate inflammation in the adipose tissue [98,150,151]. Neutrophils also secrete NE, which impairs the energy expenditure in the adipose tissue [152]) and promotes insulin resistance by degrading insulin receptor substrate 1 (IRS-1) [147]. As the number of infiltrated neutrophils augments, the activity of NE is also increased in the adipose tissue of high-fat diet mice [19,147]. Importantly, genetic deletion of NE reduces macrophage infiltration into the adipose tissue of obese mice and reverts insulin resistance [147,153], indicating that NE is a key activator of macrophages (Figure 3). An important connection between IL-1β and the NLRP3 inflammasome was found in mice fed a high-fat diet. In the adipose tissue of these mice, the mRNA levels of both IL-1β and NLRP3 were positively correlated to body weight and adiposity [154]. Furthermore, when the mice were fed a calorie-restricted diet, the mRNA levels of both molecules were significantly decreased [154]. Together, these results suggest that the interaction of neutrophils with adipocytes induces IL-1β expression via the NF-κB pathway and that free fatty acids released after lipolysis of adipocytes also stimulate neutrophils to produce high levels of IL-1β via the inflammasome pathway [30] (Figure 3).

Therefore, the chronic low-grade inflammation of the adipose tissue leads to the activation of neutrophils [17]. Neutrophil activation was first inferred from the observation that serum NE concentrations [26] or plasma MPO concentrations [155] were increased in obese individuals. More recently, these observations were confirmed at the cellular level. In peripheral blood leukocytes, the NE and MPO mRNA levels were found to be positively correlated to the BMI and serum triglyceride concentrations [113]. Furthermore, bariatric surgery, which leads to weight loss in patients, partially reduced neutrophil activation [26]. Another evidence of neutrophil activation in adipose tissues is the fact that leptin can delay apoptosis of mature neutrophils. The antiapoptotic properties of leptin on neutrophils involve activation by the leptin receptor of the NF-κB and MEK1/2 MAPK signaling pathways [127,156] (Figure 3). Activation of neutrophils is also detected by the altered responses neutrophils of obese patients have to various stimuli. In general, these neutrophils display elevated ROS production and release of proinflammatory cytokines [114]. The elevated ROS production observed in neutrophils from obese people has also been reported in neutrophils from obese individuals of other species, including dogs [157] and horses [158].

Another important antimicrobial function of neutrophils is phagocytosis. There are only a handful of reports describing this function in neutrophils from obese individuals. In one study, neutrophils from obese noninsulin-dysregulated horses had a significantly increased ROS production, but no changes were observed in terms of phagocytosis [158]. In another very early study, the phagocytosis and killing of *Candida albicans* by neutrophils from healthy (control) and diabetic individuals were compared. Phagocytosis occurred at similar levels in neutrophils from diabetic and control individuals [159]. However, the killing of *Candida* by diabetic neutrophils was impaired [159]. In contrast, a recent report of neutrophils from mice on a high-fat diet showed that the phagocytosis of *Klebsiella pneumonia* was reduced [160]. In most instances, independently of the level of phagocytosis reported, the killing capacity of neutrophils from obese individuals seems to be diminished. These findings agree with clinical observations consistently reporting that individuals with obesity are physiologically frail and have a higher risk of infections and mortality than normal-weight individuals [161,162]. Clearly, much work on neutrophil phagocytosis in obese individuals is needed to fully understand why the activated state of neutrophils from obese individuals does not translate into a more effective antimicrobial function.

Because activated neutrophils have the ability to increase their secretion of cytokines and chemokines, they are also described as the prime effectors of inflammatory responses [98]. As such, they are able to induce the recruitment and activation of the second wave of immune cells, including macrophages, dendritic cells, and lymphocytes [146,163]. Once at the inflamed adipose tissue, neutrophils recruit monocytes through the release of LL-37 (cathelicidin/CRAMP), azurocidin (heparin-binding protein), cathepsin G, proteinase 3 (PR3), and human neutrophil peptides 1–3 (HNP1–3) [164,165] (Figure 3). Neutrophils can then induce monocyte differentiation and macrophage polarization and activation. Lactoferrin [166], azurocidin [167], and HNP1–3 [168] can induce polarization of macrophages towards the M1 proinflammatory phenotype (Figure 3). In addition, LL-37 induces M1 macrophage polarization and release of proinflammatory cytokines [169] (Figure 3). Similarly, alarmin S100A9 also induces the release of proinflammatory cytokines from synovial macrophages [170]. Thus, neutrophil proteins contribute to inflammation intensification by promoting macrophage activation and release of proinflammatory cytokines (Figure 3).

### 5.5. Neutrophil Extracellular Traps (NETs)

As mentioned, another way neutrophils control infections is the production of neutrophil extracellular traps (NETs), which are fibers of decondensed chromatin (DNA) decorated with histones and antimicrobial proteins from neutrophil granules. NETs are formed and released by a dynamic cell death program known as NETosis. In addition to infections, NETosis can take place during noninfectious sterile inflammation, where neutrophils help repair damaged tissues. However, during persistent inflammation, NETs can aggravate the tissue damage [171,172]. Because obesity is associated with chronic systemic inflammation, it is possible that NETosis is activated, and NETs may contribute to some of the medical complications associated with obesity.

The role of NETs in obesity is not clear since there are conflicting reports. For example, in a diet-induced obesity mouse model, endothelial dysfunction was observed. In these mice, plasma concentrations of LL-37 were increased in mesenteric arterial walls. LL-37 was used as a marker for NETs [173]. Disruption of NETs with DNase restored the endothelium function, suggesting that NETs are increased in obesity and are responsible for endothelial dysfunction [173]. Similarly, in obese persons, plasma concentrations of MPO–DNA complexes (assessed by ELISA) were higher than in lean persons. Moreover, NETs concentrations correlated with the BMI [174]. In addition, recent bioinformatics studies found a strong relationship between obesity, inflammatory markers, such as TNF-α, IL-6, IL-8, heat shock protein 90 (HSP90), and NETs formation [175,176]. Moreover, it was also found that exercise reduces NETs [175]. Together, these reports suggest that obesity-induced inflammation is associated with elevated NETs formation. However, in other studies, an opposite relationship was reported. Using purified neutrophils from obese individuals and in vitro testing, it was found that although neutrophils displayed an activated phenotype (elevated ROS production), they exhibited lower NETs formation than neutrophils from lean individuals [114]. Similarly, using intravital microscopy in mice kept on a high-fat diet, it was revealed that neutrophils produce fewer NETs in liver vasculature than neutrophils from lean mice (kept on a normal diet) [177].

Extending these observations to neutrophils from diabetic individuals, it is also found that the role of NETs in this obesity-related condition is, again, not clear. Detecting NE–DNA complexes as an indicator of NETs, it was reported that recently diagnosed T2D patients had higher plasma levels of NETs than healthy (control) individuals [178]. Furthermore, measuring NE and histone/DNA complexes in serum, it was concluded that NETs were increased in patients with diabetic retinopathy [179]. In another report, neutrophils from diabetic patients with proliferating retinopathy also presented increased NETs production, particularly when exposed to high levels of glucose [180]. These results are similar to those reported in a previous study, where diabetic patients had elevated NETs components in serum, and neutrophils presented enhanced NETosis in case of high levels (25 mM) of glucose [181]. Together, these reports suggest that diabetic conditions, particularly high glucose, lead to enhanced NETosis. However, other reports indicate that high glucose concentrations decrease the formation of NETs [182,183].

Nevertheless, the role for NETs in diabetic wound healing seems more clearly established [184,185,186]. Diabetic patients frequently have foot lesions that do not heal. These individuals are referred to as diabetic foot patients [187]. In a model of wound healing, skin wounds were inflicted on mice. Healing of these lesions was longer in diabetic mice than in normoglycemic (control) mice [184]. Furthermore, when the wounds were treated with DNase 1 to degrade NETs, wound healing was improved both in the diabetic mice and the control mice [184]. In addition, the wounds of diabetic animals presented larger amounts of citrullinated histone H3 (H3Cit), a marker for NETosis. In contrast, no H3Cit was observed in the wounds from the *Padi4*^−/−^-mice (deficient in enzyme peptidylarginine deiminase 4 (PAD4) that causes histone citrullination), despite many neutrophils present. More importantly, the wounds of the *Padi4*^−/−^-mice healed very fast [184]. In another study, neutrophils were stimulated with PMA to induce NETosis, and NETs formation was abolished by treatment with hydrogen sulfide (H_2_S) [186]. Furthermore, diabetic mice with wounds were treated intraperitoneally with H_2_S. The wounds in these mice had decreased NETs markers (NE, MPO, H3Cit, and PAD4) and healed better than the wounds in the control mice [185]. In a more recent study, intravital microscopy detected enrichment of NETs components in the bed of excisional wounds, and inhibition of PAD4 with BB-Cl-amidine improved wound healing in diabetic mice [186]. Together, these results suggested that diabetes slows down wound healing by activating NETosis. Mechanistically, NETs slow down wound healing in diabetic animals by triggering NLRP3 inflammasome activation via the TLR-4/TLR-9/NF-κB signaling pathway in macrophages. As a result, macrophages release IL-1β and prolong the inflammatory response in the diabetic wound [188,189].

At the present time, it is not possible to decide with confidence whether NETs formation is enhanced or reduced in obesity. The discrepancy among the various studies may be related to the different methodological approaches used to evaluate NETs. In some studies, indirect assessments were made by detecting some NETs components in plasma or serum. In other studies, NETs formation was evaluated directly in vitro with purified neutrophils. Yet, in other studies, intravital microscopy was used to detect NETs. Methods detecting NETs components do not necessarily confirm that NETosis took place. Elevated circulating DNA or neutrophil granule proteins may be caused by several other reasons besides NETosis. In vitro assays with purified neutrophils are more reliable to detect NETs. In either case, authors should be aware of the limitations of the methodology used and take them into consideration when interpreting the experimental results. Another possible reason for the discrepancy among the reported results is that neutrophil function may be affected by the metabolic and inflammatory states of the individual. Earlier, it was shown that NETs formation is dependent on glucose. Upon PMA stimulation, neutrophils increased glucose uptake and their glycolysis rate (as measured with a Seahorse analyzer). In the absence of glucose, PMA induced neutrophils to decondense chromatin, but they did not release NETs. However, if glucose was added at this time, NETs release took place within minutes [190]. Based on these data, the authors suggested that NETs formation could be metabolically divided into two phases: the first, independent from exogenous glucose (chromatin decondensation), and the second (NETs release), dependent on exogenous glucose and glycolysis [190]. More recently, it was reported that neutrophils from mice fed a normal diet used glycolysis for NETs release in both physiological and inflammatory (sepsis) conditions. However, neutrophils from mice fed a high-fat diet could not release NETs after a secondary ex vivo activation despite the high glycolytic potential and the flexibility to oxidize fatty acids [191]. Thus, the metabolic and inflammation states of the individual can influence the neutrophil function. In the future, extensive well-controlled studies will be required to elucidate the role of NETs in obesity.

## 6. Neutrophils in Type 1 Diabetes (T1D)

Diabetes mellitus, commonly referred to only as diabetes, is a group of metabolic disorders characterized by hyperglycemia over long periods of time. Diabetes is due to either lack of insulin secretion from the β cells in the pancreas or insulin resistance, a condition in which cells of the body do not respond properly to insulin [192]. There are two main types of diabetes: type 1 (insulin deficiency) and type 2 (insulin resistance) [193]. Both diabetes types are associated with serious clinical complications such as cardiovascular disorders, heart failure, atherosclerosis, diabetic neuropathy, diabetic retinopathy, and diabetic kidney disease [194,195,196].

Type 1 diabetes mellitus (T1D) is considered a T cell-mediated autoimmune disease, in which autoreactive T lymphocytes destroy the insulin-producing β cells in the pancreatic islets [197]. Much progress has been made in understanding T1D thanks to the nonobese diabetic (NOD) mouse animal model [198]. Similar to human T1D, NOD mice exhibit an autoimmune response towards β cells, resulting in the dysfunction and destruction of these cells. However, a limitation of this model is that in NOD mice, the initial antigen is insulin [199], while in humans, anti-islet autoantibodies are the most frequently detected autoantibodies [200]. In a study of Japanese T1D patients, it was reported that the main antigens recognized by autoantibodies were glutamic acid decarboxylase (GAD), insulinoma-associated antigen-2 (IA-2), zinc transporter 8 (ZnT8), and insulin [201].

Pioneering work with NOD mice demonstrated that physiological β cell death induced the recruitment and activation of B-1a cells and plasmacytoid dendritic cells to the pancreas [202]. These events represent the initial immunological steps for developing T1D. Importantly, in this first study, a significant early infiltration of neutrophils to the pancreas was also reported [202]. At the same time, it was found that circulating neutrophil numbers were reduced in T1D patients and that neutrophils were present in the pancreas of patients with T1D but not in patients with T2D or in nondiabetic controls [203]. A reduction in circulating neutrophils has been confirmed in other studies and it is considered a hallmark of T1D [204,205,206,207,208]. Moreover, this reduction of neutrophils correlates with lower serum levels of NE and PR3 [209] and with faster disease progression [206,210]. The reduction of circulating neutrophils is due mainly to neutrophil infiltration into the pancreatic tissue [203,206]. In neonatal NOD mice, neutrophil infiltration [211,212] and NE concentrations in the pancreas [211,213] are already higher than in control mice as early as at two weeks of age. Macrophages and β cells produce chemokines CXCL1 and CXCL2, which in turn recruit CXCR2-expressing neutrophils to the pancreas [212] (Figure 4). Thus, neutrophils emerge as important cells participating in the early stages of T1D development. This is not too surprising, since recently neutrophils have been recognized as key components of both the innate and adaptive immune systems [105] and as important participants in the immunization and the effector phases of autoimmune diseases [214].

As seen in several autoimmune diseases, NETosis might contribute to promoting both inflammation and tissue damage. In the case of T1D, NETs components have been detected in circulation. However, there are contradictory reports. Increased NETs components (NE and PR3 proteins) were reported in the serum of patients with T1D [206,215]. Yet, in a previous report, reduced circulating levels of NETs components were found to correlate with the reduced number of circulating neutrophils [209]. However, NETosis within the pancreas clearly contributes to disease progression [216]. Neutrophil infiltration into pancreatic islets of NOD mice correlates with higher levels of citrullination [217]. Thus, by inhibiting NE with sivelestat or elafin [211] or PAD4 with BB-Cl-amidine [217], the development of diabetes was prevented in NOD mice. Similarly, by degrading NETs with staphylococcal nuclease (SNase) (delivered to NOD mice by oral administration of modified *Lactococcus lactis*), pancreatic inflammation was reduced, β cell numbers increased, and glucose tolerance was improved [218]. Moreover, neutrophils isolated from T1D patients had an increased expression of PAD4 and showed enhanced NETosis after stimulation [184]. In vitro, NETs isolated from T1D pediatric patients induced monocyte-derived dendritic cell activation, leading to the production of interferon gamma (IFN-γ) by T cells [219] (Figure 4). Although NETs do not seem to directly induce the production of autoantibodies, they favor β cell damage resulting in exposure of the antigens recognized by anti-islet autoantibodies.

Neutrophils can also directly activate B cells via secreted cytokines such as BAFF (B cell-activating factor of the TNF family). BAFF, acting through its receptor [220], is one of the main prosurvival factors for B cells as well as for antibody-producing plasma cells [221] (Figure 4). Hence, neutrophils and NET components have an evident contribution to the development of T1D. These findings may open new opportunities for innovative therapeutic approaches in the future.

In addition, several neutrophil functions, including phagocytosis, degranulation, and production of ROS, have been reported to be reduced in patients with T1D [159,222,223]. All these defects are thought to be caused by hyperglycemia [224,225]. However, it was recently found that in vitro neutrophil migration was impaired in neutrophils from T1D but not from T2D patients [207]. This functional defect was associated with the expression of L-selectin (CD62L) but not with high glucose concentrations [207]. Thus, it may be possible that certain neutrophil defects are specific features of T1D and not a general glucose-dependent defect. Future studies should look more carefully into neutrophil functions at different stages of diabetes.

## 7. Neutrophils in Type 2 Diabetes (T2D)

Type 2 diabetes mellitus (T2D) is a chronic disease characterized by an elevated concentration of glucose in blood as a result of limited insulin secretion and/or insulin resistance. As a consequence, in T2D, the metabolism of carbohydrates, lipids, and proteins is dysregulated [226].

T2D is associated with obesity-induced chronic systemic inflammation [13,15]. As described above, neutrophil infiltration into adipose tissues contributes to the development of insulin resistance. In mice fed a high-fat diet, neutrophils in the adipose tissue release NE which degrades IRS1, resulting in impaired insulin signaling [147] (Figure 5).

Therefore, genetically NE-deficient mice showed reduced adipose tissue inflammation and increased glucose tolerance, including better insulin sensitivity [147,152]. Furthermore, activated neutrophils from diabetic patients released more IL-1, IL-6, IL-8, and TNF-α than neutrophils from healthy individuals [227], leading to the increased level of circulating inflammatory cytokines. These elevated cytokines may then impact multiple organs in the body. One cytokine that has been repeatedly implicated in T2D is TNF-α [119,228]. Neutrophils from T2D patients secrete higher amounts of IL-6 and TNF-α in response to lipopolysaccharide (LPS) stimulation, resulting in insulin resistance, which then increases the blood glucose concentration [227]. Similarly, in the serum of obese patients with cardiovascular disease, larger TNF-α concentrations have been reported [229].

TNF-α has also been implicated in other mechanisms that contribute to the development of T2D, for example, altered function of endothelial cells [230,231]. Changes in the expression of adhesion molecules by vascular endothelial cells are observed in patients with T2D, and these changes seem to occur even before the onset of T2D [232]. Altered adhesion function of endothelial cells has also been associated with the progression of atherosclerosis [233]. TNF-α seems to be responsible for these alterations by inducing an increased low-density lipoprotein uptake in vascular endothelial cells [234] (Figure 5). This process can then promote atherosclerosis and extend inflammation [234]. Consequently, genetically TNF-α-deficient mice show less endothelial cell dysfunction in diabetes animal models [231,235]. In addition, TNF-α has been linked to β cell dysfunction and insulin resistance. TNF-α and IL-1 induced β cell dedifferentiation in cultured human and mouse pancreatic islets by downregulating transcription factor Fox01, which regulates β cell proliferation [236,237] (Figure 5).

The elevated levels of circulating inflammatory cytokines found in diabetic patients and animals also have important effects on neutrophil function. Neutrophils of diabetic individuals display lower phagocytic activity [159], lower production of ROS [225], and lower chemotactic capacity [238] than neutrophils from healthy control individuals. Some of these functions (migration and bacteria killing) also seem to be compromised in hyperglycemia, and can be induced in vitro upon exposure of neutrophils to serum from diabetic patients [182]. Finally, recent experiments showed that neutrophils can release microvesicles, which are involved in cell–cell communication. The neutrophil microvesicles concentration increased in the mice fed a high-fat diet. These microvesicles also accumulate in certain regions of arteries and promote vascular inflammation and atherosclerosis. In vitro, neutrophil microvesicles promoted inflammatory gene expression by endothelial cells [239]. Together, these reports suggest that neutrophils actively contribute to maintaining systemic inflammation and originating some pathological consequences found in T2D.

## 8. Concluding Ideas

Obesity is a growing health problem of pandemic proportions. Obese people develop many pathological conditions, collectively referred to as obesity-related complications, including, among others, diabetes mellitus, cardiovascular diseases, and cancer. These conditions are thought to be initiated or worsened by the mild, chronic, systemic inflammation characteristic of obesity. This inflammatory condition is initiated by dysfunctional adipocytes, but later it is perpetuated by cells of the innate immune system, primarily neutrophils and macrophages. In recent years, it has become apparent that neutrophils are the first immune cells infiltrating the obese adipose tissue. Neutrophils then get activated and release multiple inflammatory factors that recruit other immune cells and further promote inflammation. The cellular and molecular mechanisms used by neutrophils to orchestrate this scenario are only partially described. Consequently, several roles neutrophils play in obesity remain unidentified. Some of the more pressing issues that should be addressed in future research are mentioned next.

An evident characteristic of obesity is that circulating neutrophils are elevated. This increase in neutrophil blood counts seems to be associated with the severity of obesity, i.e., with the BMI and with inflammation markers such as plasma CRP. However, almost nothing is known about the relationship between neutrophil numbers and the inflammation state of persons with different grades of obesity. Using the inflammation biomarker NLR may be a way to detect an ongoing subclinical inflammation in individuals who otherwise appear healthy. This may be particularly relevant in overweight individuals who may be more susceptible to becoming obese. Although the NLR is firmly established, its use in routine clinical practice remains very limited.

We described that it is now clear neutrophils are recruited to obese adipose tissue. However, neutrophil infiltration is not the same in the adipose tissue from different individuals. What are the signals that attract neutrophils to this tissue under different conditions? The molecular nature of lipid chemotactic factors is not yet known. Future studies will help elucidate these chemotactic factors and the mechanisms they use to recruit neutrophils into adipose tissues. In general, it seems that neutrophils in obese individuals have an activated phenotype, revealed by an elevated ROS production. However, almost nothing is known about other neutrophil functions, such as chemotaxis, phagocytosis, and NETs formation. This is a very relevant issue because obese individuals appear to have a higher risk of infections and mortality than normal-weight individuals. How is it possible that “activated” neutrophils in obese people fail to control infections? Clearly, future research should concentrate on neutrophil phagocytosis in obese individuals to fully understand why neutrophils from obese individuals do not show an effective antimicrobial activity. In these future studies, it will also be important to evaluate neutrophil phagocytosis in cells from people with different grades of obesity; and to separate obesity studies from diabetes studies. Although diabetes is a complication of obesity, neutrophils from two individuals with similar obesity, one with diabetes and another without it, will most likely behave differently.

Another important topic that nowadays persists as very confusing is the role of NETs in obesity, particularly of how NETs may perpetuate inflammation and how NETs influence the antimicrobial function. As described above, presently, it is not clear whether NETs formation is enhanced or reduced in obesity. The discrepancy among the various studies seems to be related to the different methodological approaches used to evaluate NETs. Methods detecting NETs components do not necessarily confirm NETosis took place. Hence, it is imperative to validate the presence of actual NETs not only by detecting the “NETs markers” (NE, MPO, H3Cit), but also by demonstrating that actual chromatin fibers decorated with these molecules are present. This is particularly relevant when trying to detect NETs in circulating blood. In vitro assays with purified neutrophils are more reliable for detecting NETs. However, even in this case, proper confirmation that NETs are present is required. Therefore, future studies should use more than just one method to detect NETs formation. Presence of NETs can alter various aspects of cell behavior and thus generate different pathological outcomes. It is of great importance to discover what NETs do in different obesity conditions.

Neutrophil functions may be affected by the metabolic state of the individual. Reports on how hyperglycemia modify neutrophil responses are contradictory. Future studies on neutrophil responses should take into consideration the metabolic condition of the person donating the cells, and also tests in different laboratories should use similar conditions to assay neutrophils in vitro. For example, including similar glucose concentrations in all experiments.

Neutrophils are very flexible cells and, clearly, they can display multiple phenotypes [240]. Although not discussed in this review, a subpopulation of neutrophils, the so-called low-density neutrophils (LDNs), have been reported in many pathological conditions. No information on the presence or functionality of this subpopulation of neutrophils exists for obese individuals. The possible role of LDNs in obesity is also an interesting line of future research. Similarly, the influence of the microbiota on neutrophil functions is another topic not much studied [134,241]. Because microbiota composition is changed during obesity, it will be of much interest to explore how the microbiota affects neutrophil functions during obesity.

## Figures and Tables

**Figure 1 cells-11-01883-f001:**
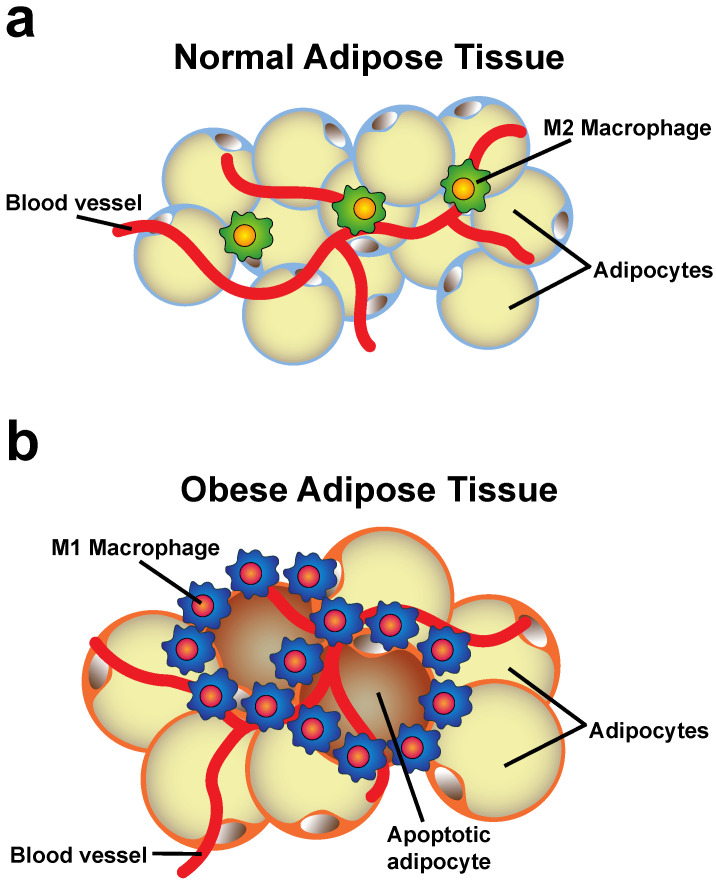
Changes in the adipose tissue. (**a**) In normal adipose tissue, adipocytes store limited amounts of fat (triglycerides). Some anti-inflammatory M2 macrophages are distributed throughout the tissue. (**b**) In obese adipose tissue, adipocytes accumulate excessive amounts of fat and become larger and stressed. These adipocytes then become dysfunctional, releasing proinflammatory adipokines that recruit and activate more macrophages into the tissue. Macrophage activation results in an increase in macrophages with the M1 (proinflammatory) phenotype and a decrease in M2 (anti-inflammatory) macrophages. Due to hypoxia and stress, adipocytes also become apoptotic. Macrophages concentrate around apoptotic and dead adipocytes, forming crown-like structures where defective adipocytes are eliminated by phagocytosis.

**Figure 2 cells-11-01883-f002:**
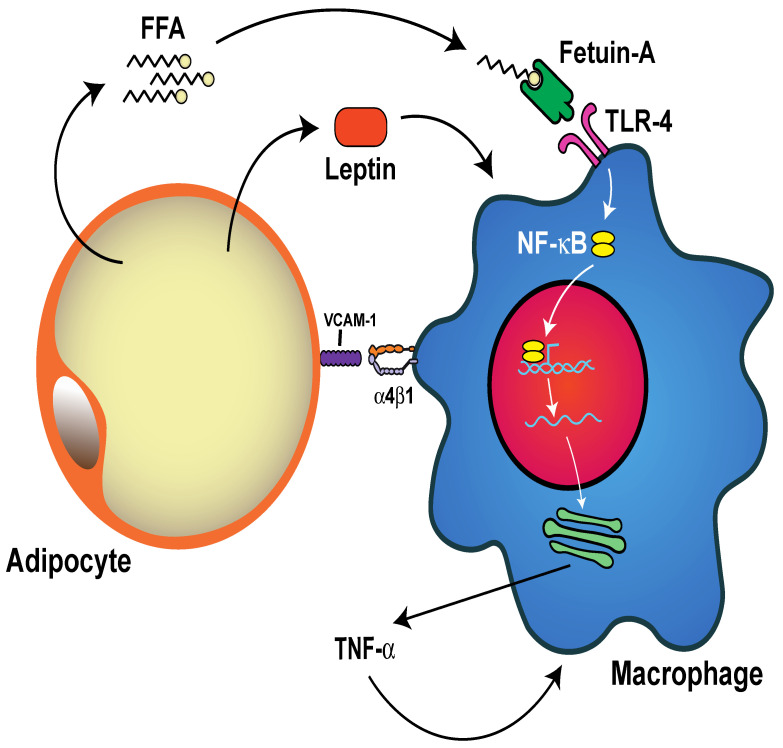
Interaction of adipocytes and macrophages in obese adipose tissue. Macrophages enter in direct contact with adipocytes through adhesion of integrin α4β1 on macrophages with vascular cell adhesion molecule-1 (VCAM-1) on adipocytes. Free fatty acids (FFA) produced by lipolysis are released by adipocytes and bind to fetuin-A, which is then recognized by toll-like receptor 4 (TLR-4) on macrophages. This leads to the activation of proinflammatory signals like the nuclear factor kappa B (NF-κB) pathway that induces secretion of cytokines such as tumor necrosis factor alpha (TNF-α) by macrophages. The leptin released by adipocytes also induces macrophages to produce TNF-α. Further activation of macrophages by TNF-α creates a self-sustained inflammation in obese adipose tissue.

**Figure 3 cells-11-01883-f003:**
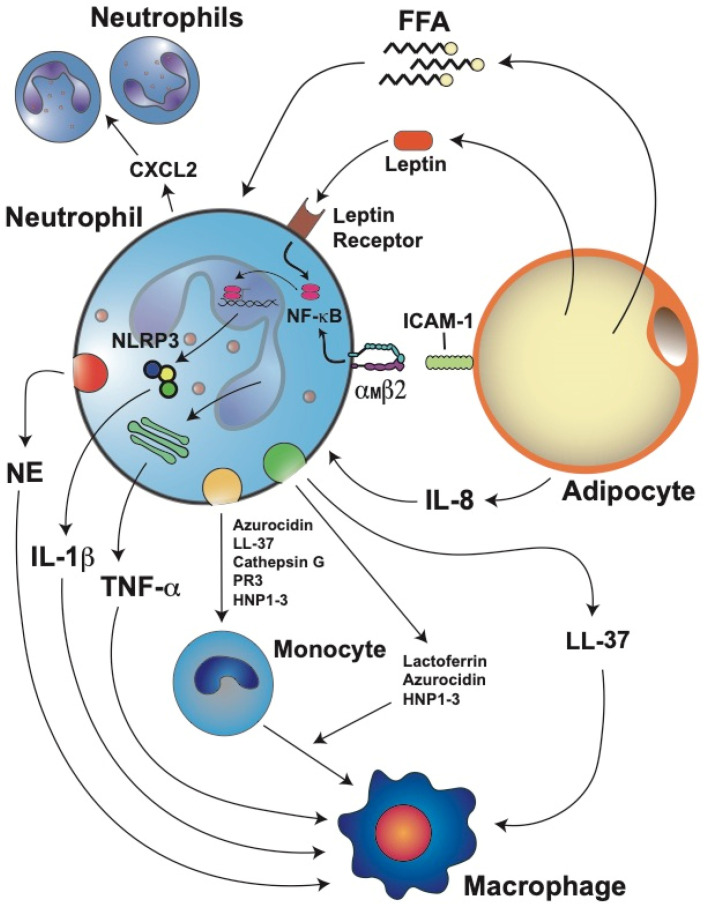
Stressed adipocytes in obese adipose tissue recruit neutrophils, which then further promote inflammation. Adipocytes produce adipokines such as leptin and cytokines such as interleukin-8 (IL-8). IL-8 is a potent chemoattractant for neutrophils. Once in the adipose tissue, neutrophils can recruit more blood neutrophils by releasing C–X–C motif chemokine ligand 2 (CXCL2), another important neutrophil chemoattractant. Neutrophils directly interact with adipocytes via the binding of integrin α_M_β2 on the neutrophil to intercellular adhesion molecule 1 (ICAM-1) on the adipocyte. This interaction activates neutrophils and induces them to produce interleukin 1 beta (IL-1β) via the nuclear factor kappa B (NF-κB) and inflammasome (NLRP3) pathways. IL-1β is an important activator of macrophages. Neutrophils also produce tumor necrosis factor alpha (TNF-α), which further stimulates macrophages. Leptin, through its receptor, also activates the NF-κB pathway, resulting in the inhibition of neutrophil apoptosis. Free fatty acids (FFA) derived from adipocyte lipolysis can also attract neutrophils and stimulate them to produce more IL-1β. Neutrophils also produce elastase (NE) which impairs energy expenditure in the adipose tissue and directly activates macrophages. Granule protein cathelicidin (LL-37) can also activate the release of more proinflammatory cytokines from macrophages. Activated neutrophils can also recruit monocytes through the release of azurocidin, LL-37, cathepsin G, proteinase 3 (PR3), and human neutrophil peptides 1–3 (HNP1–3). In addition, lactoferrin, azurocidin, and HNP1–3 can induce polarization of macrophages towards the M1 proinflammatory phenotype.

**Figure 4 cells-11-01883-f004:**
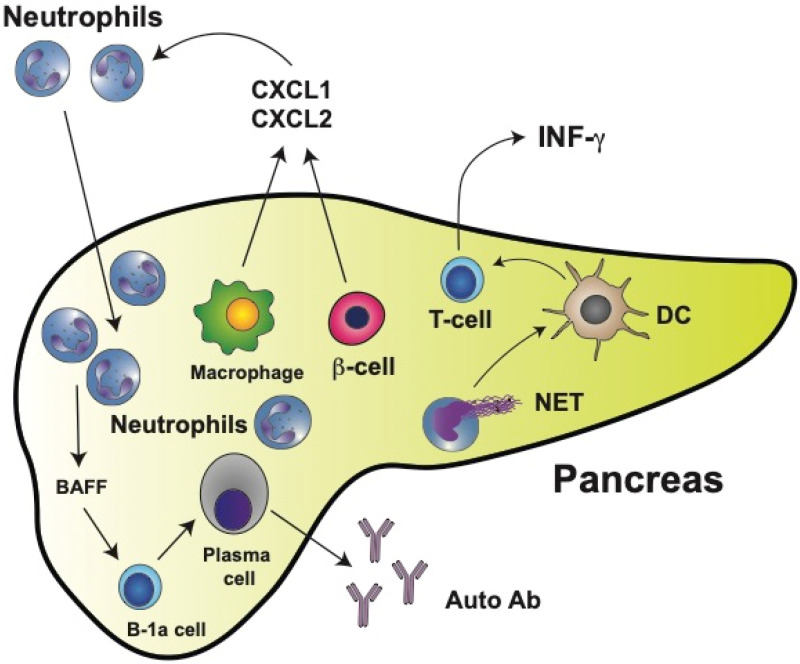
Type 1 diabetes mellitus (T1D) is characterized by neutrophil infiltration in the pancreas. In the pancreas, macrophages and insulin-producing β cells release chemokines CXCL1 and CXCL2, which in turn recruit neutrophils to the pancreas. Neutrophils can then directly activate B-1a cells via secreted cytokines such as BAFF (B cell-activating factor of the TNF family). BAFF is one of the main prosurvival factors for B cells as well as for antibody-producing plasma cells. Most plasma cells activated this way in the pancreas produce autoreactive antibodies (Auto Ab). The main antigens recognized by autoantibodies are glutamic acid decarboxylase (GAD), insulinoma-associated antigen-2 (IA-2), zinc transporter 8 (ZnT8), and insulin. Many neutrophils also release neutrophil extracellular traps (NET), which can activate dendritic cells (DCs). Activated DCs then stimulate T cells, leading to the production of interferon gamma (IFN-γ).

**Figure 5 cells-11-01883-f005:**
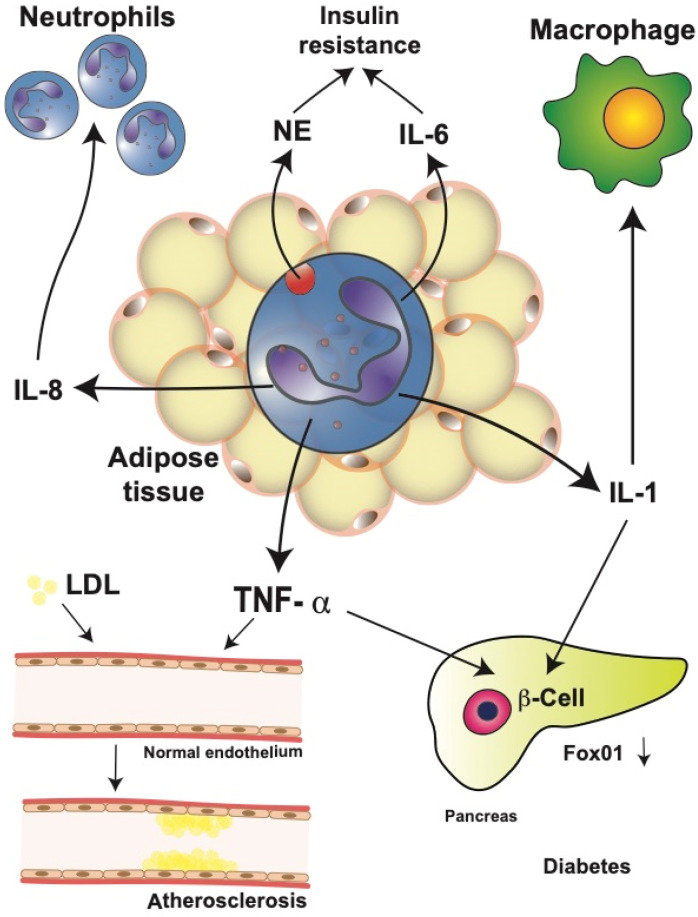
Neutrophils in obesity-related complications. Neutrophils in obese adipose tissue release large amounts of interleukin (IL)-1β, IL-6, IL-8, and tumor necrosis factor alpha (TNF-α). These cytokines have important systemic effects leading to obesity-related complications. IL-8 recruits more neutrophils into the adipose tissue, creating an amplification cycle. Neutrophil elastase (NE) and IL-6 contribute to the development of insulin resistance by impairing insulin signaling. IL-1β is an important activator of macrophages in multiple parts of the body. Furthermore, IL-1β, together with TNF-α in pancreatic islets, induces β cell dedifferentiation by downregulating transcription factor Fox01, which regulates β cell proliferation. Together, these events may result in type 2 diabetes mellitus. In addition, TNF-α can alter the adhesion function of endothelial cells by inducing increased low-density lipoprotein (LDL) uptake. These changes have been associated with the progression of atherosclerosis.

## Data Availability

This study did not report any data.

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
