# Peer review of "Neutrophils Actively Contribute to Obesity-Associated Inflammation and Pathological Complications"

_cells, 2022, doi:10.3390/cells11121883_

Round 1

Reviewer 1 Report

Uribe-Querol and Rosales, provide a review about the potential role of neutrophils on diabetes. I have found this review a bit difficult to follow. Long paragraphs about prevalence of disorders such as T1D and T2D, make the reader to loose focus. In the same way, only sections 5 and 6 seem to be related to the objective of this review. Authors should restrict the review to the proposed topic. I appreciate an introduction for the non-specialised reader should be given, but not to the extent provided by the authors. An important aspect not covered by the authors is the influence of the gut microbiota in inflammation of adipose tissue, modulating the behaviour of immune cells etc. There is a huge amount of works that have shown an active role of the gut microbiota and its metabolites in obesity and diabetes.  I strongly believe authors should dedicate a section  to cover this topic in the context of this review.

Author Response

We thank the reviewer for careful reading of our manuscript and for the helpful suggestions. We have revised the manuscript accordingly to make the introduction more focused and shorter. Also, a new section on the influence of gut microbiota in inflammation has been included. The new section 3 "Microbiota and obesity" begins in line 103 of the revised manuscript.

Reviewer 2 Report

Dear Authors,

I enjoyed reading your manuscript.

Yours sincerely,

Author Response

Dear reviewer,

   Thank you very much for your comment. We are delighted that you found our review useful.

Reviewer 3 Report

This review article is well-organized; however, this manuscript seems to be a chapter of textbook that state overview of the established achievements.

Major points.

#1. In figure 4, auto-reactive antibodies are used to describe Type 1 diabetes mellitus. Please specify the auto-antigens.

#2. In association with auto-reactive antibodies, the presentation of auto-antigen to T-cell by macrophage or dendritic cell is critical. Are NETs important antigens for pancreatic isletitis ?

Author Response

We thank the reviewer for suggesting to reduce the overview of established achievements. We have revised the manuscript accordingly to make the introduction more focused and shorter.

   We also thank the reviewer for suggesting to include information of auto-antigens in T1D. We have included information on the antigens reported to be recognized by autoantibodies. The revised manuscript now reads (line 569) " while in humans, anti-islet autoantibodies are the most frequently detected autoantibodies [200]. In a study of Japanese T1D patients, it was reported that the main antigens recognized by autoantibodies were glutamic acid decarboxylase (GAD), insulinoma-associated antigen-2 (IA-2), zinc transporter 8 (ZnT8), and insulin [201]."

   We also included some comments to address the question whether NETs are antigens for pancreatic isletitis. As far as we know, NET antigens do not generate autoantibodies. However, NET promote beta-cell damage and thus islet antigens get exposed. This idea is presented in the revised manuscript (line 609) "Although, NET do not seem to directly induce the production of autoantibodies, they favor β-cell damage resulting in exposure of antigens recognized by anti-islet autoantibodies."

Round 2

Reviewer 1 Report

The authors have reasonably addressed all the comments I made.